# Design and Performance Analysis of a Micro-Displacement Worktable Based on Flexure Hinges

**DOI:** 10.3390/mi13040518

**Published:** 2022-03-26

**Authors:** Lan Yan, Anna Jiang, Feng Jiang, Guangda Liu, Fuzeng Wang, Xian Wu

**Affiliations:** 1College of Mechanical Engineering and Automation, National Huaqiao University, Xiamen 361021, China; yanlan@hqu.edu.cn (L.Y.); 21013080015@stu.hqu.edu.cn (A.J.); xianwu@hqu.edu.cn (X.W.); 2Institute of Manufacturing Engineering, National Huaqiao University, Xiamen 361021, China; wangfuzeng1987@163.com; 3Xiamen Xiazhi Technology Tool Co., Ltd., Xiamen 361006, China; zhangjinxian@xiatec.com

**Keywords:** flexure hinge, micro-displacement, four degrees of freedom, finite element simulation

## Abstract

The flexure hinge is a kind of micro-displacement adjustment device with application prospects because of its high displacement resolution, positioning accuracy and repeatability. In this study, a micro-displacement worktable with four degrees of freedom ($\overset{\to }{X}$, $\overset{\to }{Z}$, $\overset{︵}{X}$, $\overset{︵}{Z}$) was designed. The micro-displacement worktable was composed of three different flexure hinges. The adjustment ranges and adjustment accuracy of flexure hinges in terms of their respective degrees were improved. The micro-displacement worktable performance was examined by FEA (Finite Element Method). The maximum displacement that was adjusted in $\overset{\to }{X}$ and $\overset{\to }{Z}$ was 1.67 µm and 1.74 µm. The maximum angle adjusted in the $\overset{︵}{X}$ and $\overset{︵}{Z}$ direction was 14.90° and 18.58°. A test platform was developed for micro-displacement worktable performance tests. The simulation results showed a good agreement with the experimental results.

## 1. Introduction

In the field of micro-manufacturing [1], precision measurement [2,3], medical instruments [4,5] and friction tester [6], the performance of the ultra-precision positioning table are more and more important. In recent years, the micro displacement mechanism based on the principle of flexure hinge has been widely developed. The mechanism has characteristics of compact structure, free of mechanical friction and high displacement resolution [7]. 

The original flexure hinge is a single structure that owns only one degree of freedom, and its adjustment range is limited. With the development of flexure hinges, it was gradually evolved from a single degree of freedom to multiple degrees of freedom. Paros [8] firstly introduced a theoretical approach to analyzing flexure hinges in 1965. Lobontiu not only discussed the rotation ability and motion accuracy of the right-angle flexure hinge but also proposed a theoretical analysis of multi-axis compatibility with the flexure hinge assembly process [9,10,11]. In addition, he proposed a flexure hinge calculation method of conical, curved surfaces, including elliptical and parabolic linear. Wu [12] improved the design formula of the flexibility or spring stiffness of rectangular single-axis hinges and right-circle hinges based on Paro’s theory. Lin [13] recently designed and analyzed the performance of a hybrid flexure hinge. Pan [14] introduced the triple periodic minimum surface lattice structure and designed a new type of leaf flexible hinge. Kong [15] proposed a new type of conical V-shaped flexure hinges (CFHs). Wang [16] designed high-rate, low-stress, non-friction and non-bending flexure hinges for bridge compliant mechanisms. Qiu [17] proposed a variable pitch folding flexible hinge (PFFH) based on the theory and characteristics of coil spring. Li [18] proposed two generalized models for fast solving the flexibility and precision closed-form equations of multi-axis flexure hinges with small deflections.

With the improvement of the flexure hinge theory, more and more research has focused on flexure hinge assembly for multi-degree of freedom adjustments. Sun [19] and Li [20] designed a displacement-amplified gripper to achieve displacement adjustment of 2-DOF on the plane. Shao [21] presented a 2-DOF PFSM with a cross-axis decoupling capability. Qin [22] and Lin [23] designed 3-DOF flexure hinges that increase the rotation of the z-axis. A 3-DOF platform, which improved stability, was designed by Lee [24]. Zhao [25] designed a 4-DOF micro-displacement adjustment mechanism to achieve three directions of movement and one direction of rotation. Chen [26] described a 5-DOF alignment to compensate for alignment errors. With the continuous development of robot technology, flexure hinge structures of multiple degrees of freedom were used in the design of robots. Hesselbach [27] and Yun [28] designed a 6-DOF parallel robot using a flexure hinge. The 6-DOF platform improved structural stability and had better dynamic mechanical properties. Lu [29] designed a new three-dimensional ultrasonic vibration platform with adjustable characteristics based on T-shaped flexible hinges. Gräser [30] proposed a high-precision compliant 2-DOF micro-positioning platform with flexible hinges, which can realize the motion range of the X–Y axis ±10 mm. Gan [31] proposed a design of a 3-DOF bidirectional motion platform based on a Z-shaped flexible hinge and bridge mechanism. Gui [32] designed a three-degree-of-freedom flexible hinge micro-motion platform based on a new composite lever amplification mechanism, which has good motion decoupling and high precision. Flexure hinges play an increasingly important role in the precision instrument.

Finite element simulation plays an important role in the design and performance check of mechanical parts. Therefore more people are using FEA to design flexure hinges [33]. Cai [34] and Ivanov [35] used a simulation to analyze the mechanical properties of trapezoidal flexure hinges and rounded flexure hinges. Ma [36] presented a miniature piezoelectric-driven fatigue device with 3-DOF. Hwang [37] theoretically analyzed the working characteristics of flexure hinges and lever mechanisms. The safety of a piezoelectric ceramic actuator device based on a flexure hinge was checked by finite element simulation [38]. The flexibility of this kind of flexure hinge finger was greatly improved. 

However, most of the reported flexure hinges have a small range of displacement, which cannot meet the application fields with higher accuracy requirements. For example, in the field of MEMS technology, when processing, positioning and assembling small parts, most need to adjust the position of small parts from multiple perspectives. In addition, flexure hinge structures that can adjust multiple degrees of freedom are most complicated, which has the problems of a large number of components; a heavy weight; long assembly time; and inter-mechanism friction, wear and lubrication. Thus, to solve the above problems, this paper designed a novel 4-DOF flexure hinge micro-displacement worktable, which is composed of three different flexure hinges. The worktable has the advantages of 4-DOF, high stiffness and good decoupling effect, and obtains a large range of micro-displacement adjustments to achieve high-precision displacement output. It can be applied to the field of precision machining, precision measurement, microelectronic manufacturing, and so on. Furthermore, the micro-displacement worktable has the advantages of simple structure, small volume, no mechanical friction, no gap and high motion sensitivity. The performance of the worktable was examined by finite element analysis. A test platform was developed for micro-displacement worktable performance tests. The micro-displacement worktable was tested by experiments, and the experimental results were compared with the simulation results. The experiment result shows that the flexure hinge has the potential for micro-manufacturing and precision measurement, which helps to improve the accuracy and reliability of the micro-displacement worktable. The results of the micro-displacement table are compared with other similar designs, as shown in Table 1.

## 2. Design of Flexure Hinge Mechanisms

The micro-displacement working platform can improve micro-cutting precision, so it is suitable for micromachining and precision measurement. The micro-displacement working platform is shown in Figure 1. The platform includes flexure hinges A, B and C. The flexure hinge A consists of a parallel four-bar mechanism, which can realize horizontal and vertical displacement adjustment. The flexure hinges B and C can achieve angular displacement adjustment. The most widely used flexure hinge in this micro-displacement working is corner-filleted. The structure diagram of the corner-filleted is shown in Figure 2a. The force condition of the flexure hinge is shown in Figure 2b.

When the left end is subjected to the load $F_{y}$, the displacement at the free end (*x* = 0, *y* = 0) is expressed as follows.
(1)$$\mu_{1}=0$$
(2)$$\;\omega_{1}=\frac{F_{y}l^{3}}{3EI}+\frac{\left(1+\mu \right)}{4}\frac{F_{y}lt^{2}}{EI}$$

When the left end is subjected to the bending moment $\;M_{Z}$, the displacement at the free end (*x* = 0, *y* = 0) is expressed as follows.
(3)$$\mu_{2}=0$$
(4)$$\omega_{2}=-\frac{M_{Z}l^{2}}{2EI}$$

When the left end is subjected to the load $F_{X}$, the displacement at the free end (*x* = 0, *y* = 0) is expressed as follows.
(5)$$\mu_{3}=\frac{F_{X}l}{Ebt}.\;$$

The total displacement of the flexure hinge is shown in Equation (6):(6)$$\left\{\begin{array}{c} \mu_{i}=\mu_{1}+\mu_{2}+\mu_{3}=\frac{F_{X}l}{Ebt}\\ \omega_{i}=\omega_{1}+\omega_{2}=\frac{F_{y}l^{3}}{3EI}+\frac{3F_{y}l}{2Gbt}-\frac{M_{Z}l^{2}}{2EI}\\ \;\theta_{i}=\theta_{1}+\theta_{2}=-\frac{F_{y}l^{2}}{2EI}+\frac{M_{Z}l^{2}}{2EI} \end{array}\right.$$

When the displacement of both ends of the flexure hinge is not zero, the overall stiffness of the flexure hinge under combined deformation is as shown in Equations (7) and (8). The displacement range of the platform can be calculated easily by using the angular displacement matrix.
(7)$$\left\{\begin{array}{c} \;\;F=k\delta \\ F={\left[F_{Xi}\;\;\;F_{yi}\;\;\;\;M_{Zi}\;\;\;\;F_{Xj}\;\;\;\;F_{yj}\;\;\;\;\;M_{Zj}\right]}^{T}\\ \;\;\;\delta ={\left[\mu_{i}\;\;\;\;\omega_{i}\;\;\;\;\theta_{i}\;\;\;\;\mu_{j}\;\;\;\;\omega_{j}\;\;\;\;\theta_{j}\right]}^{T} \end{array}\right.$$
(8)$$k=\left[\begin{array}{cc} \begin{array}{ccc} \frac{Ebt}{l} & 0 & 0\\ 0 & \frac{12EIGbt}{Gbtl^{3}+18EIL} & \frac{6EIGbt}{Gbtl^{2}+18EI}\\ 0 & \frac{6EIGbt}{Gbtl^{2}+18EIL} & \frac{2EI\left(2Gbtl^{2}+9EI\right)}{Gbtl^{3}+18EI} \end{array} & \begin{array}{ccc} -\frac{Ebt}{l} & 0 & 0\\ 0 & -\frac{12EIGbt}{Gbtl^{3}+18EIL} & \frac{6EIGbt}{Gbtl^{2}+18EI}\\ 0 & -\frac{6EIGbt}{Gbtl^{2}+18EI} & \frac{2EI\left(2Gbtl^{2}+9EI\right)}{Gbtl^{3}+18EI} \end{array}\\ \begin{array}{ccc} -\frac{Ebt}{l} & 0 & 0\\ 0 & -\frac{12EIGbt}{Gbtl^{3}+18EIL} & -\frac{6EIGbt}{Gbtl^{2}+18EI}\\ 0 & \frac{6EIGbt}{Gbtl^{2}+18EIL} & \frac{2EI\left(2Gbtl^{2}+9EI\right)}{Gbtl^{3}+18EI} \end{array} & \begin{array}{ccc} \frac{Ebt}{l} & 0 & 0\\ 0 & \frac{12EIGbt}{Gbtl^{3}+18EIL} & -\frac{6EIGbt}{Gbtl^{2}+18EI}\\ 0 & -\frac{6EIGbt}{Gbtl^{2}+18EI} & \frac{2EI\left(2Gbtl^{2}+9EI\right)}{Gbtl^{3}+18EI} \end{array} \end{array}\right]$$

When input $\overset{\to }{X}$ and $\overset{\to }{Z}$ direction displacement of hinge A at the point A and B occurs in Figure 3a, the displacement of the $\overset{\to }{X}$ and $\overset{\to }{Z}$ direction is adjusted. When input Y displacement of flexure hinge B at point C in Figure 4a and flexure hinge C at point D in Figure 5a occurs, the displacements of the $\overset{︵}{X}$ and $\overset{︵}{Z}$ direction are adjusted. When flexure hinges A, B and C are assembled, the linear adjustment of hinge A and angular displacement adjustment of hinge B and C can be combined. The micro-displacement can achieve 4-DOF micro-displacement adjustment in precision machining confidential measurement.

The input displacement $\overset{\to }{X}$ and $\overset{\to }{Z}$ direction are applied following the arrow in Figure 3. The middle part of the flexure hinge is no response area. When the input displacement of the $\overset{\to }{X}$ direction at point A and $\overset{\to }{Z}$ direction at point B is changed, the output displacement of the Y direction of the supporting surface is not affected. The input displacements of the $\overset{\to }{X}$ and $\overset{\to }{Z}$ can not only be loaded at the same time, but it can also be loaded separately, and thus the positioning accuracy of the $\overset{\to }{X}$ and $\overset{\to }{Z}$ can be ensured. The displacement of output in the X direction is $l_{3}$ and that in the Z direction is $h_{8}$.
(9)$$\tan\theta_{1}=\frac{l}{h}$$
(10)$$l_{1}=\tan\theta_{1}*h_{1}\;$$
(11)$$\theta_{2}=\theta_{3}$$
(12)$$\tan\theta_{2}=\frac{l_{1}}{h_{3}}$$
(13)$$l_{1}=\tan\theta_{2}*h_{3}$$
(14)$$l_{3}=\tan\theta_{3}*h_{4}=\frac{lh_{1}h_{4}}{hh_{3}}$$
(15)$$\theta_{4}=\theta_{5}$$
(16)$$\tan\theta_{4}=\tan\theta_{5}=\frac{l_{4}}{h_{5}}=\frac{l_{5}}{h_{6}}$$
(17)$$h_{8}=h_{6}=h_{7}=\frac{h_{5}l_{5}}{l_{4}}$$

As shown in Figure 4, the input Y-direction displacement of the flexure hinge B at point C is applied at the marked position, and the output response surface can realize angular displacement output in the $\overset{\to }{X}$ direction. The output angle of $\overset{\to }{X}$ direction is $\theta_{1}.$
(18)$$\tan\theta_{1}=\frac{l}{h}=\frac{l_{1}}{h_{1}}$$

As shown in Figure 5, the input Y-direction displacement of the flexure hinge C at point D is applied at the marked position, and the output response surface can realize the angular displacement output in the $\overset{︵}{Z}$ direction. The output angle of $\overset{︵}{Z}$ direction is $\theta_{2}.$
(19)$$\tan\theta_{1}=\tan\theta_{2}=\frac{l}{h}=\frac{l_{2}}{h_{1}}$$

## 3. Finite Element Analysis

In order to investigate the maximum output displacement of flexure hinges and examine the performance of flexure hinges, the FEM (Finite Element Method) model of flexure hinges was established. AISI 1045 steel was selected as the material of the model. The physical properties of AISI 1045 steel are shown in Table 2 (http://www.matweb.com, accessed on 10 January 2022) [39]. The flexure hinge was meshed by triangular mesh with a mesh size of 0.1 mm. The meshes are refined at flexure hinges to guarantee simulation accuracy. The input displacement points and fixed points of the flexure hinge A, B and C are shown in Figure 3, Figure 4 and Figure 5, respectively. Then flexure hinges A, B and C were assembled in the FEM model.

Figure 6 illustrates the corresponding stress distribution acting on flexure hinges during the movement. As shown in Figure 6, the stress concentration occurs at flexure hinges. The stress of flexure hinges must be less than the allowable stress (407.7 MPa) of the material. Figure 7 shows the relationship between the stress of flexure hinges and the input displacement. From this Figure 7, it can be seen that the maximum input displacement of $\overset{\to }{X}$ and $\overset{\to }{Z}$ direction of the flexure hinge A is 8 µm. The maximum input displacement of hinge B and C is 12 µm and 18 µm, respectively. Flexure hinges work safely when the input displacement is within the specified range of displacement. If the input displacement is beyond the displacement range, flexure hinges may be damaged.

The output displacements of three flexure hinges are calculated using FEA. When the stress of the danger points is less than allowable stress, it can be seen input displacement and output displacement have linear relationships. The results are shown in Figure 8 and Figure 9. The maximum displacement adjusted in $\overset{\to }{X}$ and $\overset{\to }{Z}$ direction of the flexure hinge A is 1.73 µm and 1.88 µm. The maximum angle in $\overset{︵}{X}$ and $\overset{︵}{Z}$ direction of hinge B and hinge C is 15.1° and 22.3°. Because the assembly flexure hinges are more flexible than a single flexible hinge, the coupling is not serious, the range of movement and rotation is larger and there are different output displacements. For the assembly of these three parts, the output displacements of response surfaces of assembly flexure hinges are calculated. The maximum displacement adjusted in the $\overset{\to }{X}$ and $\overset{\to }{Z}$ direction of the flexure hinge A is 1.67 µm and 1.74 µm. The maximum angle in the $\overset{︵}{X}$ and
$\overset{︵}{Z}$ and direction of hinge B and hinge C is 14.90° and 18.58°. Table 3 summarizes all simulation data of the flexure hinge.

## 4. Experiment Tests

Experimental tests are carried out to investigate flexure hinges input displacement and adjustable range. The test platform is shown in Figure 10. The displacement of the flexure hinge is at the micrometer level, so the tiny vibration can cause a serious error in the experimental results. A vibration-isolated optical platform (ZDT18-10) was used as the base of the test platform. The amplitude of this floating platform was 0.002 mm. The vibration-isolated optical platform can isolate the outside disturbance from the experiments. A linear motion actuator whose positioning accuracy is 0.05 µm was used as a loading device. The displacements on the output terminals are recorded via a laser displacement sensor (LK-H080). The flexure hinge was fixed on the fixture, flexure hinges were loaded displacement by the linear motion actuator, and the test point of the laser displacement sensor was placed on the corresponding output surface for output displacement data collection.

## 5. Experimental Results

Flexure hinges were tested on the test platform. The experimental results are shown in Figure 11, Figure 12 and Figure 13. There are linear parts and nonlinear parts in the input–output curves. The material of the flexure hinge deformed plastically when the nonlinear part occurred. Therefore, the flexure hinge should work below the critical transition displacement between the linear part and the nonlinear part.

Figure 11 shows the experimental results of the $\overset{\to }{X}$ and $\overset{\to }{Z}$ direction of flexure hinge A. It can be seen that input displacement and output displacement remain liner. Flexure hinges lose function when the input displacement is beyond 8 µm. The maximum displacement in $\overset{\to }{X}$ and $\overset{\to }{Z}$ directions are 1.67 µm and 1.76 µm, respectively. The ratio of input/output displacement is 0.2195 and 0.2087, respectively. The angles of flexure hinge B and flexure hinge C are measured by an indirect measurement method in Figure 12a,b, using Equation (9) to calculate the output angle.
(20)$$\theta =\arctan\left(\frac{L}{H}\right)$$

The experimental results of the flexure hinge B and hinge C are shown in Figure 12. There is an obvious relationship between input displacement and output angle when the input displacement is lower than 12 µm and 18 µm, respectively. The maximum angle adjusted in the $\overset{︵}{X}$ and
$\overset{︵}{Z}$
is 15.1° and 22.3°, respectively. The ratio of input/output angle of hinge B and hinge C is 1.3774 and 1.2649, respectively. The simulation results show a good agreement with the simulated results.

The flexure hinge assembly was tested on the experimental platform. The experimental results for the direction of 
$\overset{\to }{X}$ and
$\overset{\to }{Z}$
of the flexure hinge A are shown in Figure 13a. From this Figure, it can be seen that the flexure hinge assembly still retains a good linear relationship, similar to the simulation results. The maximum displacement in $\overset{\to }{X}$ and $\overset{\to }{Z}$ directions are 1.67 µm and 1.74 µm, respectively. The ratio of input/output displacement is 0.2218 and 0.2217, respectively. The experimental results for $\overset{︵}{X}$ and $\overset{︵}{Z}$ displacement of the flexure hinge B and hinge C assembly are shown in Figure 13b. The maximum angle in the $\overset{︵}{X}$ and $\overset{︵}{Z}$ is 14.90° and 18.58°, respectively. The ratio of input/output angle of the hinge C is 1.3258 and 1.0643, respectively. Table 4 summarized all experimental data of flexure hinge and compared the experimental results with simulation results.

## 6. Conclusions

This paper introduced the design, finite element simulation test and experimental analysis of a new 4-DOF micro-displacement worktable. The 4-DOF micro-displacement worktable consists of three flexure hinges, which are simple and have good stability. The theoretical analysis of the designed flexure hinge was carried out, and the performance of the flexure hinge was tested by finite element simulation. A flexure hinge structure was developed by using electro-discharge wire cutting. A test platform was built to test the performance of the flexure hinge. The maximum output displacement in $\overset{\to }{X}$ and $\overset{\to }{Z}$ directions are 1.67 µm and 1.74 µm. The maximum output angle that can be adjusted in the $\overset{︵}{X}$ and $\overset{︵}{Z}$ direction is 14.90° and 18.58°. The maximum output error of the experimental and the simulated is 6.0%, 2.0%, 1.7% and 3.4%. The simulation results show a good agreement with the experimental results. This micro-displacement worktable can be used in the fields of precision machining, precision measurement, microelectronic manufacturing, etc.

## Figures and Tables

**Figure 1 micromachines-13-00518-f001:**
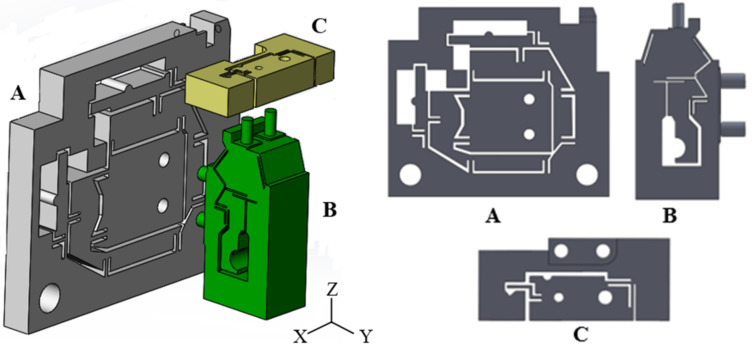
Schematic diagram of the micro-displacement working platform. The flexure hinge A can realize horizontal and vertical displacement adjustment. The flexure hinges B and C can achieve angular displacement adjustment.

**Figure 2 micromachines-13-00518-f002:**
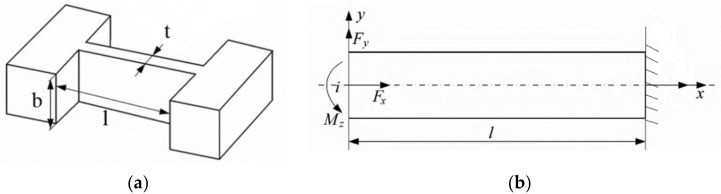
Flexure hinge. (**a**) Structure diagram of corner-filleted, (**b**) Force condition of the flexure hinge unit.

**Figure 3 micromachines-13-00518-f003:**
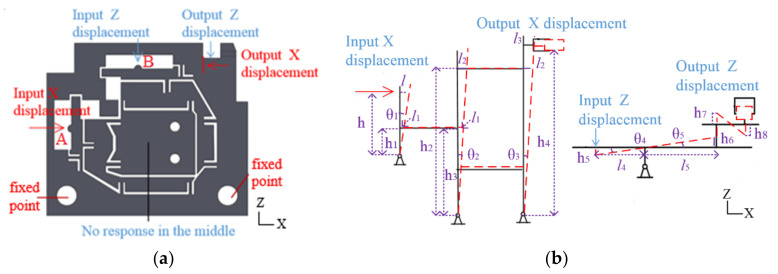
Double parallel four-bar mechanism flexure hinge A. (**a**) Flexure hinge A. (**b**) Schematic diagram.

**Figure 4 micromachines-13-00518-f004:**
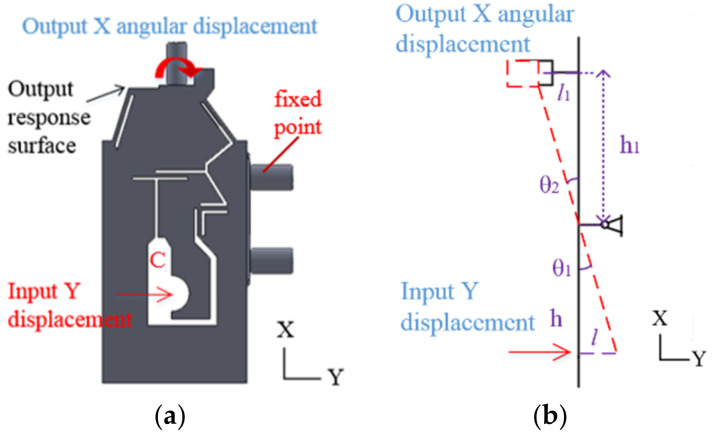
The angular displacement adjustment flexure hinge B. (**a**) Flexure hinge B. (**b**) Schematic diagram.

**Figure 5 micromachines-13-00518-f005:**
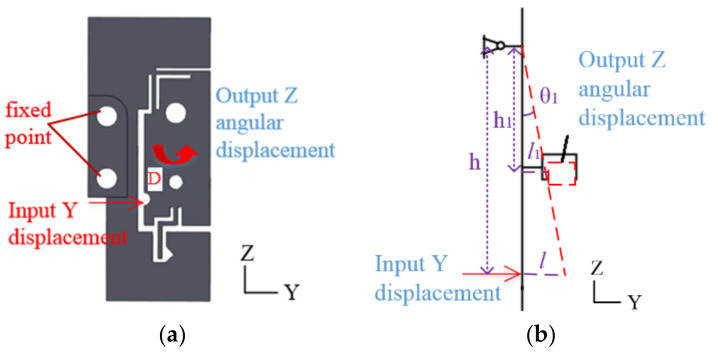
The angular displacement adjustment flexure hinge B. (**a**) Flexure hinge C. (**b**) Schematic diagram.

**Figure 6 micromachines-13-00518-f006:**
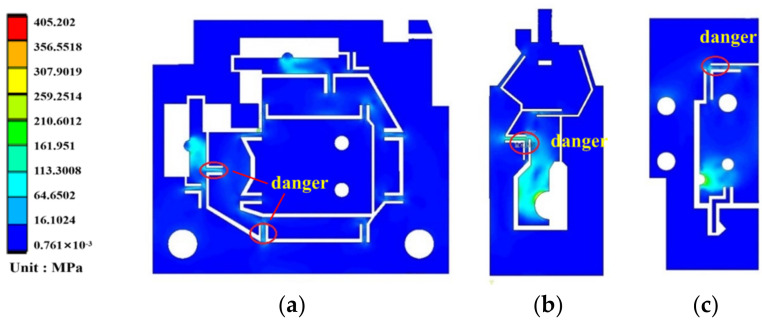
Stress analysis diagram of single flexure hinge. (**a**) Flexure hinge A. (**b**) Flexure hinge B. (**c**) Flexure hinge C.

**Figure 7 micromachines-13-00518-f007:**
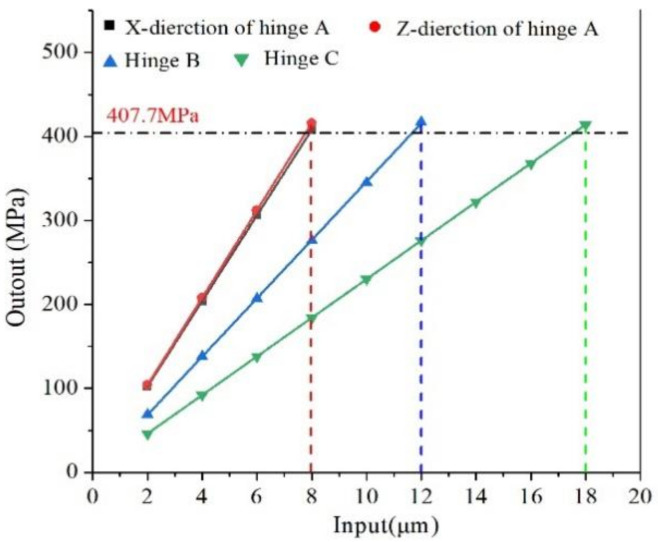
The relationship between input displacement and danger points stress.

**Figure 8 micromachines-13-00518-f008:**
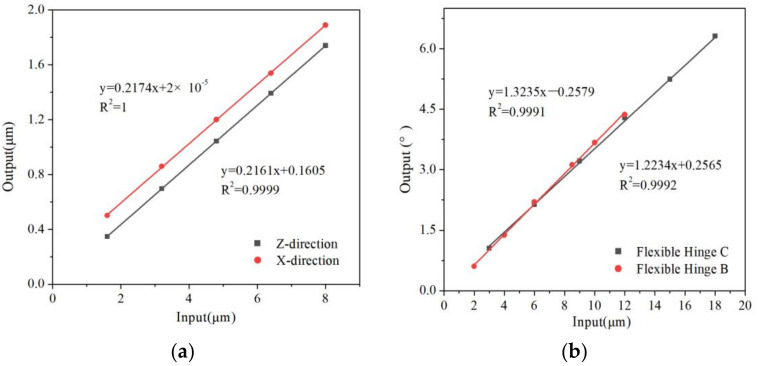
The ratio of input/output displacement of the flexure hinge. (**a**) Flexure hinge A, (**b**) Flexure hinge B and C.

**Figure 9 micromachines-13-00518-f009:**
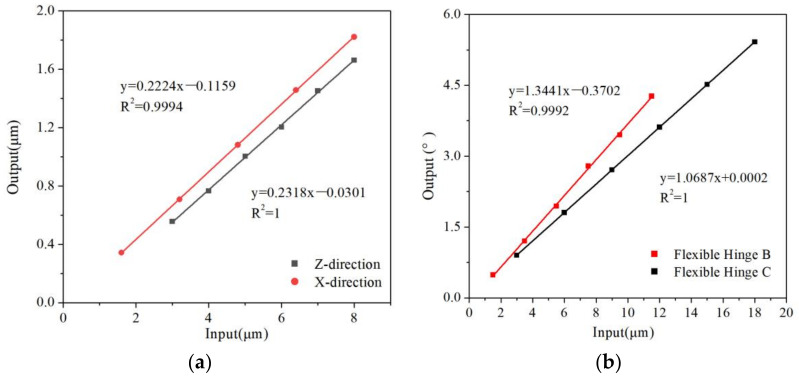
The ratio of input/output displacement of the flexure hinge assembly. (**a**) Flexure hinge A. (**b**) Flexure hinge B and C.

**Figure 10 micromachines-13-00518-f010:**
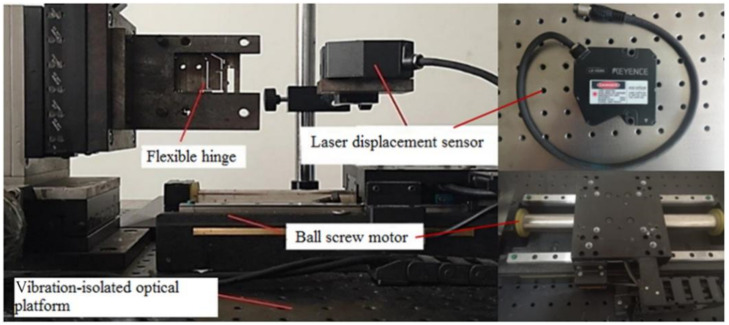
Text platform for flexure hinge.

**Figure 11 micromachines-13-00518-f011:**
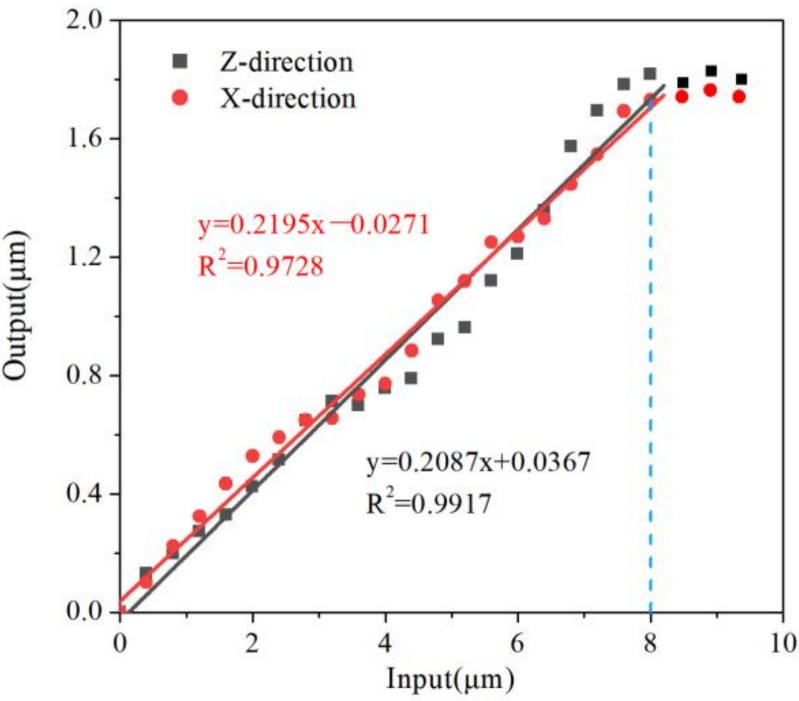
The output of flexure hinge A.

**Figure 12 micromachines-13-00518-f012:**
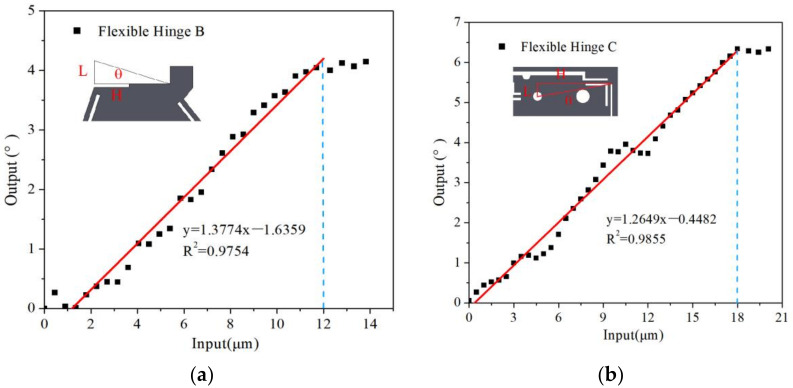
The output of flexure hinge B and C. (**a**) Flexure hinge A. (**b**) Flexure hinge B and C.

**Figure 13 micromachines-13-00518-f013:**
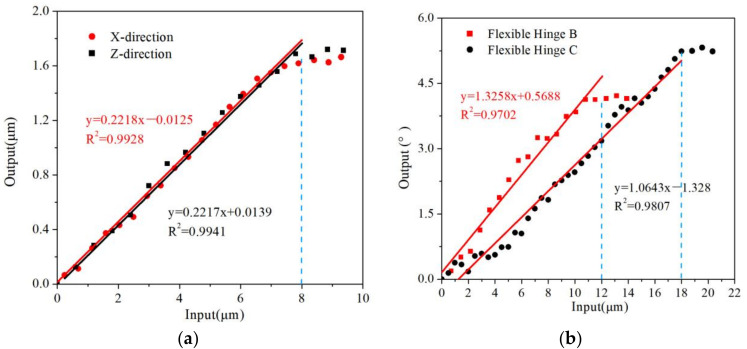
The output flexure hinge in the assembly. (**a**) Flexure hinge A. (**b**) Flexure hinge B and C.

**Table 1 micromachines-13-00518-t001:** Comparison of the results of the micro-displacement table with other similar designs.

Worktable Design	Degree of Freedom	Maximum Displacement	Maximal Angle	Maximum Error of Simulation and Test Results	Characteristics of Worktable
Yan (2022) Design and performance analysis of a micro-displacement worktable based on flexure hinges	4-DOF: X-direction Z-direction X-rotation Z-rotation	1.67 µm 1.74 µm	14.90° 18.58°	6.0%, 2.0%, 1.7%, 3.4%	The worktable has a large displacement range, high precision, simple structure and adopts a flexible hinge with a round angle.
Zhao (2007) Realization of Four-DOF Precision Adjustment Mechanism of Mirror With Flexure Hinge	4-DOF: Z-direction X-rotation Y-rotation Z-rotation	±0.4 mm	±0.08° ±0.5° ±0.06°	No simulation	The worktable has enough high resolution and adopts a single straight round flexible hinge.
Li (2017) A Compact 2-DOF Piezoelectric-Driven Platform Based on “Z-Shaped” Flexure Hinges	2-DOF: X-direction Y-direction	17.65 µm 15.45 µm	None	2.03% 3.62%	The working stress of the worktable is the smallest, and the ‘Z’ flexible hinge is adopted.
Lin (2017) Design and Performance Testing of a Novel Three-Dimensional Elliptical Vibration Turning Device	3-DOF: X-direction Y-direction Z-direction	26 µm 24 µm 22 µm	None	11.2% 11.1% 9.8%	This worktable has a compact structure with relatively large stroke and high working bandwidth, good tracking accuracy, relatively high resolution and low hysteresis, which is suitable for micro-nano processing.
Gan (2021) Design of a 3DOF XYZ Bi-Directional Motion Platform Based on Z-Shaped Flexure Hinges	3-DOF: X-direction Y-direction Z-direction	±125.58 µm ±126.37 µm ±568.45 µm	None	5.67% 3.73% 12.74%	The workbench has a compact structure, large stroke and bidirectional movement, but it is relatively difficult to manufacture, has a high cost and it is difficult to control four actuators.

**Table 2 micromachines-13-00518-t002:** The physical property of AISI 1045 steel.

Modulus of Elasticity (GPa)	Poissons Ratio	Yield Strength (MPa)	Allowable Stress (MPa)	Elongation (%)	Density (g/mm^3^)
206	0.29	530	407.7	≥12	7.85

**Table 3 micromachines-13-00518-t003:** The simulation data of flexure hinge.

Simulation of Single Flexure Hinge	Simulation (µm/°)	Simulation In/Out Ratio	Simulation of Flexure Hinge Assembly (µm/°)	Simulation In/Out Ratio
X-direction of hinge A	1.73	0.2174	0.2195	0.2174
Z-direction of hinge A	1.88	0.2161	0.2087	0.2161
Hinge B	4.37	1.3235	1.3774	1.3235
Hinge C	6.20	1.2234	1.2649	1.2234

**Table 4 micromachines-13-00518-t004:** The experimental data of flexure hinge assembly.

Experimental of Single Flexure Hinge	Experimental (µm/°)	Error (%)	Experimental of Flexure Hinge Assembly (µm/°)	Error (%)
X-direction of hinge A	1.67	3.5	1.67	6
Z-direction of hinge A	1.76	6.4	1.74	2
Hinge B	4.25	1.7	4.20	1.7
Hinge C	6.28	1.4	5.23	3.4

## Data Availability

The authors confirm that the data supporting the findings of this study are available within the article. In addition, the data that support the findings of this study are available from the corresponding author, Feng Jiang, upon reasonable request.

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
