# Peer review of "Design and Performance Analysis of a Micro-Displacement Worktable Based on Flexure Hinges"

_micromachines, 2022, doi:10.3390/mi13040518_

Round 1

Reviewer 1 Report

This paper presents the design and performance analysis of a micro-displacement worktable based on a flexure hinge, but the novelties of the presented work are not clearly described. The authors should explicitly describe the novelty.

In the introduction section, current research gaps in flexure hinges design should be clearly pointed out, namely,  the disadvantages.

More latest references should be reviewed to indicate the timeliness of the  presented work

The format for lines 100-105 should be revised.

The Engish should be improved. The grammar mistakes should be carefully checked.

In line  180, the author indicates that the maximum displacement adjusted in x- and z-direction is 1.7um and 1.88 um.  is it the displacement of Flexure hinge A? and the followed maximum displacement in X and Z 
direction is 1.67 µm and 1.74 µm, is hinge A, or B, or C? The authors should carefully check the description related to Fig.7, Fig.8, and Fig.9.

The authors should give some explanations about the difference between Figs.8 and 9. Why there are different outputs for the flexure hinge and its assembly?

Reviewer 2 Report

This paper discusses the fabrication of a high-precision worktable and is also fully analyzed and verified by scientific methods. There are several suggestions:
1. The file format needs to be adjusted, especially Equations.
2. What is the target application of this worktable, can it be compared with the design of a similar specification or the same target.
3. Discussion of the results, whether there is an additional carrier, and the comparison of the results under stress and strain.

Round 2

Reviewer 1 Report

The authors have revised the manuscript accordingly 

Author Response

Thanks again for the support and suggestions of the reviewers.

Reviewer 2 Report

The format style of the manuscript still needs to be modified.

Adding a Table to compare the worktable results with the other design of a similar specification or the same target from references would be easy to illustrate the paper's novelty.

Author Response

The authors are grateful for the reviewer’s suggestion. We have added a table to the manuscript according to the suggestion of the reviewer to illustrate the novelty of the paper.

This manuscript is a resubmission of an earlier submission. The following is a list of the peer review reports and author responses from that submission.